# Hepatic Transcriptome Profiling Reveals Lack of *Acsm3* Expression in Polydactylous Rats with High-Fat Diet-Induced Hypertriglyceridemia and Visceral Fat Accumulation

**DOI:** 10.3390/nu13051462

**Published:** 2021-04-25

**Authors:** Kristýna Junková, Lukáš F. Mirchi, Blanka Chylíková, Michaela Janků, Jan Šilhavý, Martina Hüttl, Irena Marková, Denisa Miklánková, Josef Včelák, Hana Malínská, Michal Pravenec, Ondřej Šeda, František Liška

**Affiliations:** 1Institute of Biology and Medical Genetics, First Faculty of Medicine, Charles University and General University Hospital, 128 00 Prague, Czech Republic; kristyna.junkova@lf1.cuni.cz (K.J.); lfmirchi@gmail.com (L.F.M.); blanka.chylikova@lf1.cuni.cz (B.C.); jankum@centrum.cz (M.J.); michal.pravenec@fgu.cas.cz (M.P.); ondrej.seda@lf1.cuni.cz (O.Š.); 2Department of Genetics of Model Diseases, Institute of Physiology, Czech Academy of Sciences, 142 20 Prague, Czech Republic; Jan.Silhavy@fgu.cas.cz; 3Institute for Clinical and Experimental Medicine, 140 21 Prague, Czech Republic; mabw@ikem.cz (M.H.); irma@ikem.cz (I.M.); mild@ikem.cz (D.M.); haml@ikem.cz (H.M.); 4Institute of Endocrinology, 116 94 Prague, Czech Republic; jvcelak@endo.cz

**Keywords:** metabolic syndrome, high-fat diet, insulin resistance, hypertriglyceridemia, polydactylous rat, spontaneously hypertensive rat, liver transcriptome, *Acsm3*

## Abstract

Metabolic syndrome (MetS) is an important cause of worldwide morbidity and mortality. Its complex pathogenesis includes, on the one hand, sedentary lifestyle and high caloric intake, and, on the other hand, there is a clear genetic predisposition. PD (Polydactylous rat) is an animal model of hypertriglyceridemia, insulin resistance, and obesity. To unravel the genetic and pathophysiologic background of this phenotype, we compared morphometric and metabolic parameters as well as liver transcriptomes among PD, spontaneously hypertensive rat, and Brown Norway (BN) strains fed a high-fat diet (HFD). After 4 weeks of HFD, PD rats displayed marked hypertriglyceridemia but without the expected hepatic steatosis. Moreover, the PD strain showed significant weight gain, including increased weight of retroperitoneal and epididymal fat pads, and impaired glucose tolerance. In the liver transcriptome, we found 5480 differentially expressed genes, which were enriched for pathways involved in fatty acid beta and omega oxidation, glucocorticoid metabolism, oxidative stress, complement activation, triacylglycerol and lipid droplets synthesis, focal adhesion, prostaglandin synthesis, interferon signaling, and tricarboxylic acid cycle pathways. Interestingly, the PD strain, contrary to SHR and BN rats, did not express the *Acsm3* (acyl-CoA synthetase medium-chain family member 3) gene in the liver. Together, these results suggest disturbances in fatty acid utilization as a molecular mechanism predisposing PD rats to hypertriglyceridemia and fat accumulation.

## 1. Introduction

Metabolic syndrome (MetS) is a complex of several pathophysiological conditions, including obesity, impaired glucose metabolism, hypertriglyceridemia accompanied by decreased high-density lipoprotein levels, and/or arterial hypertension [1]. MetS is a worldwide health problem and major risk factor in cardiovascular morbidity and mortality with multifactorial pathogenesis, combining the extrinsic (environment, lifestyle, diet, energy expenditure) and intrinsic (genetic background) factors and their mutual interactions. The heritability of the individual components of MetS varies from 25–80% [2].

Increasingly large genome-wide association studies (GWASs) in humans are powerful tools to uncover the genetic landscape of MetS and other multifactorial conditions and may promise prospective risk assessment in the future using polygenic risk scores [3]. However, human GWASs use very simplistic additive genetic models. As a consequence, a substantial amount of gene–gene interactions and gene–environment interaction effects on disease risk remain buried in the so-called “missing heritability” [4]. Genetically defined animal models can provide a useful alternative to GWAS in dissecting the genetic background of various polygenic traits and gene–gene and gene–environment interactions.

In the current study, we use two models of MetS, spontaneously hypertensive rat (SHR/OlaIpcv; SHR hereafter) and polydactylous rat (PD/Cub; PD hereafter) and compare them to a Brown Norway (BN) strain that is resistant to MetS development [5].

When fed a high-sucrose diet, the PD strain develops marked hypertriglyceridemia but is not susceptible to blood pressure rise, while SHR is a typical model of essential hypertension [5,6,7]. PD and SHR thus present specific subtypes of MetS. The three strains were used to derive a family of BXH/HXB recombinant inbred (RI) strains [8]. These strains represent a model system for genetic and correlation analysis of parameters of metabolic syndrome. Since the RI strains were never tested when fed a high-fat diet, our current study was performed to provide pilot data for follow-up analyses of the effects of a high-fat diet in RI strains.

The aim of our study was to unravel the genetic and pathophysiological background of higher susceptibility to the development of MetS in a PD rat strain. Polydactylous rat, like many other laboratory strains (including SHR), originated from Wistar rats [9]. It is considered as a model of MetS; however, in contrast to SHR, it was not subjected to selective breeding for high blood pressure or hypertriglyceridemia; its propensity to develop MetS was fixed by genetic drift during nonselective inbreeding.

So far, most studies in the PD model have involved dietary challenges by high-carbohydrate diets, but the effect of a high-fat diet (HFD) on MetS development in the PD strain has not been analyzed yet. Our study shows that after exposure to HFD, PD rats develop striking hypertriglyceridemia, obesity, and insulin resistance compared to both SHR and BN strains. Analysis of the liver transcriptome revealed dysregulation of multiple pathways, which can contribute to impaired lipid utilization, hypertriglyceridemia, and accumulation of fat, concomitant with decreased glucose tolerance and hyperinsulinemia. A most promising candidate gene contributing to the susceptibility of PD rats to MetS is *Acsm3* (acyl-CoA-synthetase for medium-chain member 3), which belongs to a family of enzymes activating medium-chain fatty acids (C4-C14) to beta-oxidation [10], and which was absent in the liver of PD on both mRNA and protein levels.

## 2. Materials and Methods

### 2.1. Animals

We used the following three inbred strains of Rattus norvegicus: spontaneously hypertensive rat (SHR/OlaIpcv, RGDID 61000, derived from outbred Wistar Kyoto rats by Okamoto and Aoki [11]), polydactylous rat (PD/Cub, RGDID 728161, kept since 1969 at the Institute of Biology and Medical Genetics, First Faculty of Medicine, Charles University in Prague, derived from polydactylous pair of outbred Wistar rats), and Brown Norway (BN/Cub, RGDID 60985, derived from a brown mutation in a pen-bred colony in 1958 by Billingham and Silvers [12], from the USA Institute of Biology, First Faculty of Medicine in Prague in 1964 and bred since by brother x sister mating for more than 100 generations [13].

### 2.2. Experimental Protocol

All animal experiments were conducted in compliance with the Animal Protection Law of the Czech Republic and were approved by the Ethics Committee of the First Faculty of Medicine, Charles University, Prague (Permit Number: MSMT-19427/2019-8) and Ethics Committee of the Institute of Physiology, Czech Academy of Sciences, Prague (Permit Number: 66/2014). Adult male rats of BN (*n* = 8), SHR (*n* = 6), and PD (*n* = 7) strain were used. Until the age of 6 months, they were fed a standard chow (ssniff R-Z) diet ad libitum. Then, after an overnight fasting period, we recorded the body weight and fasting glycemia, and performed the oral glucose tolerance test (OGTT). Subsequently, rats were fed HFD (Ssniff EF R/M with 30% fat, containing saturated fatty acids (FAs), short- to long-chain FAs, corresponding to approximately 50% of energy, see Appendix A) for four weeks. Rat males had free access to food (except the initial and final overnight fasting period) and water, and they were held under humidity- and temperature-controlled conditions on a 12-12-h light-dark cycle. Diet consumption was measured as the difference in weight of the feeding dose of pellets and remaining pellets after 3–4 days (twice each week). After 4 weeks (after overnight fasting period), the oral glucose tolerance test was performed again, and each male’s body weight was recorded. After, males were sacrificed by decapitation.

### 2.3. Tissue Collection

Selected tissues (liver, heart, kidneys, adrenal glands, brown fat tissue, epididymal fat tissue, retroperitoneal fat tissue, soleus muscle, and diaphragm) were dissected and weighted (except the diaphragm). The tissues were flash-frozen in liquid nitrogen and stored at −80 °C for future analysis.

### 2.4. Biochemical Parameters

#### 2.4.1. Lipidogram

Serum lipoprotein levels (including triacylglycerol (TAG) and cholesterol levels in each of 20 lipoprotein fractions) after feeding HFD were measured by LipoSearch using HPLC (high-performance liquid chromatography) [14]. Total TAG and cholesterol before and after HFD were measured using commercially available kits (Erba Lachema, Brno, Czech Republic and Roche Diagnostics, Basel, Germany). 

#### 2.4.2. FFA

Free fatty acid levels before and after feeding HFD were measured using commercially available kits (Roche Diagnostics, Basel, Germany).

#### 2.4.3. Insulin

Insulin levels before and after HFD feeding were determined using the Rat Insulin ELISA kit (Mercodia AB, Uppsala, Sweden).

#### 2.4.4. C-peptide, GIP, GLP-1, Glucagon, Leptin, PP, and PYY

The metabolic profile of rat males was assessed via Milliplex Catalog ID.RMHMAG-84K-08.Rat Metabolic Hormone MAGNETIC kit using the BioPlex 200 system (Bio-Rad Laboratories, Inc., Hercules, CA, USA, Merck Millipore Corp., Billerica, MA, USA) for levels of C-Peptide 2, GIP, GLP-1, glucagon, insulin, leptin, PP, and PYY before and after HFD. 

#### 2.4.5. Adiponectin

Adiponectin serum levels before and after HFD feeding were determined using the Rat Adiponectin ELISA kit (MyBioSource, San Diego, CA, USA). 

#### 2.4.6. Cytokine Levels

The cytokine profiles of male rats before and after HFD were assessed via Bio-Plex Pro^TM^ Rat Cytokine 23-Plex Assay (Bio-Rad Laboratories, Inc., Hercules, CA, USA, Luminex Corporation, Austin, TX, USA) for levels of G-CSF, GM-CSF, GRO/KC, IFN-γ, IL-1α, IL-1β, IL-2, IL-4, IL-5, IL-6, IL-7, IL-10, IL-12p70, IL-13, IL-17, IL-18, M-CSF, MCP-1, MIP-1α, MIP-3α, RANTES, TNF-α, and VEGF using the BioPlex system (Bio-Rad Laboratories, Inc., Hercules, CA, USA).

#### 2.4.7. Lipid Levels in Liver Tissue

To determine TAG and cholesterol in the liver, samples were extracted in a chloroform/methanol mixture. The resulting pellet was dissolved in isopropyl alcohol, after which the TAG content was determined by enzymatic assay (Erba-Lachema, Brno, Czech Republic). Hepatic diacylglycerols were separated using a modified Folche method followed by thin-layer chromatography, as previously described [15], and its quantitative content was determined by enzymatic assay (Erba-Lachema, Brno, Czech Republic).

### 2.5. Gene Expression

#### 2.5.1. RNA Isolation

Liver and epididymal fat tissue were chosen for RNA isolation due to its crucial role in lipid and lipoprotein metabolism. Tissues were homogenized in Trizol (Thermo Fisher Scientific, Waltham, MA, USA) using TissueLyser LT (Qiagen, Hilden, Germany). Chloroform was then used for phase separation, the aqueous phase was mixed 1:1 with 70% ethanol, applied to RNeasy mini spin columns, and purified using RNeasy Plus Mini Kit (Qiagen, Hilden, Germany) according to the manufacturer’s protocol. RNA was quantified by spectrophotometry and its integrity was measured using an Agilent Bioanalyzer 2100 on RNA-6000 Nano-LabChip (Agilent, Böblingen, Germany). Only samples showing RNA integrity numbers (RIN) above 8 were used for further analysis.

#### 2.5.2. Gene Expression

The transcriptome of four rats per strain (liver tissue) was assessed using Affymetrix GeneChip^®^ Rat Gene 2.1 arrays (Affymetrix, Santa Clara, CA, USA). Expression console software (Affymetrix, Santa Clara, CA, USA) was used to perform quality control. Transcriptome Analysis Console (TAC version 4.0.1, Affymetrix, Santa Clara, CA, USA) was used for data normalization, statistical analysis, and pathway enrichment using the robust multiarray average method (RMA). The microarray data discussed in this publication has been deposited in NCBI’s Gene Expression Omnibus [16] and are accessible through GEO Series accession number GSE164176.

#### 2.5.3. RT-PCR and qPCR

The different expression of candidate genes was verified by quantitative real-time PCR (qPCR). In total, 1 µg of total RNA was used to synthesize cDNA using SuperScript III reverse transcriptase (Invitrogen, Carlsbad, CA, USA) according to the manufacturer’s protocol. The resulting cDNAs were then used as a template in quantitative real-time PCR (qPCR) reactions. Primers for qPCR reactions were designed using PrimerBLAST [17] to span at least one exon–exon junction and amplification was done in 7900HT in Power-up SYBRGreen master mix (Thermo Fisher Scientific, Waltham, MA, USA). All primers are listed in Appendix A.

### 2.6. DNA Sequencing

Sequencing of *Acsm3* in PD strain: Because of the impossibility of amplifying cDNA (absence of cDNA of *Acsm3* in liver tissue from PD), genomic DNA was used for sequencing of *Acsm3*. Long-range PCR products were sequenced on an Illumina MiSeq using the Nextera XT DNA Library Preparation Kit (Illumina). Bioinformatic analysis was done with the help of Galaxy [18]. Mapping was done by BWA-MEM (Burrows-Wheeler Aligner). Duplicated reads were removed by Picard [19]. FreeBayes was used to identify sequence variants [20]. IGV (Integrated Genome Viewer) was used for data visualization. 

### 2.7. Western Blotting

Mouse monoclonal Acsm3 antibody (G-8; sc-377173) was purchased from Santa Cruz Biotechnologies, Inc., Dallas, TX, USA. Control mouse monoclonal Vinculin antibody (VLN01) was purchased from Thermo Fisher Scientific, Waltham, MA, USA. Liver lysates (3 samples from each strain) were run on SDS-PAGE (12% separating gel), and proteins were blotted onto PVDF membranes Immobilon P (EMD Millipore Biosciences, Billerica, MA, USA). Membranes were incubated overnight at 4 °C with primary antibodies at a final dilution of 1:200 (Acsm3) and 1:10,000 (Vinculin). Secondary anti-mouse antibody and signal was detected using an ECL Prime chemiluminescent detection kit and Hyperfilm ECL (all from GE Healthcare Bio-Sciences, Chicago, IL, USA). Developed hyperfilms were scanned and densitometry was performed in ImageJ [21]. 

### 2.8. Statistical Analysis

Statistical evaluation of morphometric data and metabolic parameters was performed using STATISTICA 12 (TIBCO Software, Palo Alto, CA, USA). Data are presented as arithmetic means ± SEM if not indicated otherwise. The groups were compared by using ANOVA (analysis of variance–tissue weight, insulin levels, gene expression in selected tissues), ANOVA for repeated measurements (oGTT, AUC before and after exposure to HFD, animal weight), and general linear model ANOVA (lipoprotein fractions, with 3 factors–strain, cholesterol or TAG, fraction) if the variances within each group were similar. In case of different variances within the group, the nonparametric test (Kruskal–Wallis test) was used. The comparison was followed by the post-hoc Tukey test. The cutoff for the significant results was determined as a *p*-value lower than 0.05. In the case of gene expression data (lined up by their *p*-value), false discovery rate (FDR) correction was conducted, applying the Benjamini–Hochberg procedure, where α = 0.05 (5 % FDR). The FDR procedure was repeated for post-hoc comparison of each of the two strains, and a fold change cutoff <−1.2 or >1.2 was applied. For the qPCR experiment, the cycle threshold (Ct) values of selected genes were normalized relative to the expression of the peptidylprolyl isomerase A (*Ppia*) (cyclophilin) gene (for liver tissue) and ribosomal protein L41 (*Rpl41*) gene (for epididymal fat tissue), which served as the internal control, with results being determined in triplicates. Relative quantification was performed using the ΔΔCt method. Statistical analysis was performed using ANOVA.

### 2.9. Gene Enrichment Analysis and Functional Clustering of the Differentially Expressed Genes

Annotation of the differentially expressed genes was performed using DAVID v6.7 software (The Database for Annotation, Visualization and Integrated Discovery) to explore the functions of all of the differentially expressed genes [22,23]. Using default settings on DAVID software, differentially expressed transcripts were compared to the whole transcriptome background using χ^2^, Fisher’s exact test, binomial probability, and hyper-geometric distribution to obtain an enrichment score, which is the geometric mean of all enrichment *p*-values. Thus, hierarchical categories based on the molecular function (MF), cellular component (CC), and biological process (BP) features were created. A Venn diagram of the differentially expressed genes among the strains was created using BioVenn [24].

## 3. Results

### 3.1. Morphometric Profile and Oral Glucose Tolerance Test

No significant differences in glycemia (fasting and all measurements during OGTT) were found among the three strains before feeding HFD (Figure 1a). During the HFD administration, there was a significant weight gain in the PD strain compared to the other strains and SHR compared to BN (Figure 1b,c), reflecting the amount of the consumed diet (Figure 1d). After feeding HFD, the OGTT showed impaired glucose clearance in the PD strain compared to the other strains (Figure 1e,f). The relative weights of fat depots in all locations (retroperitoneal, epididymal, and brown) were significantly highest in the PD strain compared to the other strains, corresponding to the weight gain. Relative heart and kidney weights were significantly higher in SHR compared to the other strains. The weight of liver/100 g body weight was significantly higher in PD and SHR than in BN (Figure 1g).

### 3.2. Metabolic Profile

Although no significant strain differences of insulinemia were seen before HFD, after feeding HFD, the insulin levels increased significantly in the PD strain compared to the other strains (Figure 2a). After HFD, the cholesterol levels increased in chylomicrons and very low-density lipoprotein (VLDL) particles (fractions 1–5, i.e., particles with diameter >44.5 nm) of the PD strain. The increase was relatively modest, reaching statistical significance in ANOVA but not in post-hoc comparison of individual groups (Appendix A). On the other hand, there was an order of magnitude increase of total TAG levels due to their accumulation in chylomicron and VLDL fractions in PD compared to the other strains after HFD (Figure 2b,c). No significant differences in total cholesterol and TAG levels were found before HFD. After HFD, we found a significant decrease in the FFA level in PD and SHR, but PD compared to SHR and BN still manifested the significantly highest FFA level (Figure 2d). No significant differences in the adiponectin level before and after HFD were found (Appendix A). The serum leptin level before HFD feeding was significantly higher in PD compared to the other strains. However, after feeding HFD, we found no significant difference between the PD and BN strain and only subtly higher leptinemia in PD compared to SHR. These data indicate that the leptin level does not correspond to the amount of fat tissue (after correction to the white adipose tissue amount, represented by the sum of retroperitoneal and epididymal fat, we found a significantly lower leptin level in PD compared to the other strains, Figure 2e,f). No significant differences among the three strains were found in cytokine levels (MCP-1, IL-1a, IL-1b, IL-2, IL-4, IL-5, IL-6, IL-7, IL-10, IL-12, IL-13, IL-17 IL-18, G-CSF, GM-CSF, GRO/KC, IFN-γ, M-CSF, MIP-1a, MIP-3a, RANTES, TNF-α, VEGF) before and after HFD (Appendix A) or in C-peptide, glucagon, GIP, GLP-1, or PP levels (Appendix A).

### 3.3. Lipid Levels in Liver Tissue after HFD Feeding

As shown in Figure 2g, comparison of the cholesterol content of liver tissue did not show any significant difference between PD and other strains. Hepatic TAG levels were the highest in SHR compared to PD but not significant compared to BN. On the other hand, thin-layer chromatography assessment of diacylglycerols (DAG), lipotoxic intermediates of TAG metabolism, revealed a significantly lower amount of DAG in SHR liver compared to the other two strains (Figure 2g).

### 3.4. Gene Enrichment Analysis in Liver Tissue 

After multiple comparison correction using the Hochberg–Benjamini procedure at a significance level of 0.05, we found in total 5480 differentially expressed genes. In each post-hoc pairwise comparison of strains, we found 2636 for BN vs. PD (1064 upregulated in PD, 1572 downregulated in PD), 1906 for BN vs. SHR (936 downregulated in SHR, 970 upregulated in SHR), and 938 for PD vs. SHR (620 downregulated in PD, 318 upregulated in PD) (Figure 3a). The complete lists of the differentially expressed genes are provided in Appendix A.

### 3.5. Pathway Analysis 

Transcriptome analysis for the PD vs. SHR differentially expressed genes indicated significant enrichment of the pathways related to tryptophan metabolism, beta oxidation, retinol metabolism, and fatty acid biosynthesis (Figure 3b). Interestingly, the same pathways were also the most significantly enriched for the SHR vs. BN comparison (Figure 3c), albeit with a different mix of enriched genes. In the PD vs. BN differentially expressed genes pathway enrichment analysis, the highest scoring results were rhodopsin-like G-protein-coupled receptors, proteasome degradation, fatty acid biosynthesis, cytoplasmic ribosomal proteins, and glucuronidation (Figure 3d). Interestingly, in both SHR vs. PD and SHR vs. BN comparisons, SHR displayed overexpression of complement activation, focal adhesion, and interferon signaling, and none of these pathways were significantly enriched in PD vs. BN comparison.

### 3.6. qPCR

We confirmed differential expression for 13 protein-coding genes with connection to MetS traits development using qRT-PCR. After finding that *Acsm3* (acylCoA-synthetase for medium-chain family member 3) hepatic expression was absent in the PD strain, we also measured the expression of three other closely related acyl-CoA-synthetases, which were not differentially expressed in the microarray data. The most substantial difference in expression was seen in *Scd1* (stearoyl-CoA-desaturase 1, Figure 4a), which was overexpressed in PD compared to the other strains (26-fold and 16-fold, compared to BN and SHR, respectively). We noticed significant downregulation in PD of genes involved in activation of FAs–*Acsm3* (acyl-CoA-synthetase medium-chain family member 3), *Acsm2a* (acyl-CoA-synthetase medium-chain family member 2A), especially the *Acsm3* gene, which was completely/nearly completely absent, as shown by failure of amplification by standard RT-PCR techniques (Figure 4a and Figure 5a). The expression levels of the remaining genes are shown in Figure 4a.

Since many of the genes differentially expressed in the liver exhibit a close connection to lipid metabolism, we measured the expression of selected genes (*Acsl5*, *Acsm2a*, *Acsm3*, *Casp12*, *Rgs16*, *Scd1*, *Sirt3*) in epididymal fat tissue using qPCR. We found a significantly higher expression of *Scd1* in PD compared to other strains; no other significant result was found. *Acsm3* and *Acsm2a* were practically not expressed in epididymal fat tissue (Figure 4b). Because we observed deregulated leptin levels in the PD strain, we determined Lep gene expression in epididymal fat tissue, which correlated with fat tissue mass in the strains (Figure 4c).

### 3.7. Sequencing of Acsm3

The absence of *Acsm3* expression in the liver of the PD strain compared to SHR and BN strains (Figure 5a) suggests a sequence variant in a tissue-specific cis-regulatory element (e.g., promoter, enhancer, differential splice site) precluding liver expression. However, sequencing of genomic DNA of the PD strain did not reveal any causal mutation in the *Acsm3* coding sequence, introns, and core promoter, which might explain the absence of transcript and ACSM3 protein. 

### 3.8. Western Blot Analysis

The absence of *Acsm3* liver expression in PD on the mRNA level prompted us to test the expression of the protein. Western blot analysis of liver tissue lysates proved the absence of ACSM3 protein in PD (Figure 5b).

## 4. Discussion

It is known that PD rats develop marked hypertriglyceridemia after being fed a high-sucrose diet [6]; however, the effects of a high-fat diet on the development of MetS have not been thoroughly tested yet. For the experiment and subsequent comparison, two other control rat strains were chosen, SHR and BN. SHR is also considered as a model of MetS: after high-sucrose diet feeding, rats of SHR develop MetS traits (including visceral obesity, lipid and glucose metabolism impairment), characterized by constitutive arterial hypertension [8]. The BN strain, on the other hand, does not develop MetS [5,6,7].

Six-month-old PD male rats fed standard chow had most metabolic and morphometric parameters indistinguishable from SHR and BN except for higher body weight. However, 4 weeks of HFD had a disproportionally strong impact on PD rats, leading to manifestation of MetS traits, such as obesity, hypertriglyceridemia, impaired oral glucose tolerance, and hyperinsulinemia. 

The most pronounced difference was markedly elevated TAG levels in PD rats compared to SHR. Interestingly, this hypertriglyceridemia was not accompanied by hepatic steatosis. This seemingly contradictory observation might be due to increased triglyceride release from the liver of PD rats as is suggested by the nearly 10 times higher VLDL triglycerides in the plasma of PD rats when compared to SHR and BN rats (Figure 2b). This mechanism can be supported by upregulation of one of the lipogenesis genes in liver transcriptome—*Scd1*, encoding the stearoyl-CoA-desaturase 1 enzyme. Higher expression of *Scd1* is possibly compensatory due to a high FAs supply and could be accompanied by higher VLDL output in PD. Alternatively, increased VLDL levels can be due to decreased lipolysis, as chylomicron levels were also excessively high. In addition, significantly higher diacylglycerol (DAG) levels were found in the liver of both PD and BN strains compared to SHR. It is not clear whether increased DAG levels predispose PD rats to insulin resistance [25,26].

Liver transcriptome pathway analysis showed significant downregulation of focal adhesion, adipogenesis, triacylglycerol synthesis, and lipid droplet metabolism pathways in PD in relation to SHR. We also found that pathways associated with chronic or acute inflammatory processes-prostaglandin synthesis, interferon signaling, and complement activation were uniquely downregulated in PD in comparison to the SHR strain. Several studies showed the important role of innate immune activation as a key factor in triggering and amplifying hepatic inflammation in NAFLD/NASH [27]. Other papers described proinflammatory fatty liver disease induced by overnutrition-triggered lipotoxicity [28]. However, there was no immune activation in PD as indicated by the serum cytokine levels. We can only speculate whether long-term HFD feeding would not lead to more dramatic differentiation in systemic inflammation among the strains.

Another finding potentially relevant for MetS pathogenesis in PD was a significantly lower relative leptin serum level (after correction to fat tissue weight). After HFD, the absolute serum leptin level was highest in PD, but supposing a linear correlation between the serum leptin level and fat tissue amount, our data showed a deficient serum leptin level in PD when corrected to the fat tissue amount. This suggests impairment of the leptin pathway after HFD. A similar finding was described previously [29] and based on this fact, it is possible that after longer-term HFD feeding, leptin levels could become elevated, and correlate with fat reserves. However, since there could be insufficient signal provided to the hypothalamus in PD about energy reserves, it could fail to prevent higher energy intake, which we indeed observed in PD in the 4-week HFD feeding course compared to the other strains that exhibited a plateau phase, after an initial weight increase. Leptin gene expression in epididymal fat corresponded to the fat tissue amount in all strains. The putative defect is thus more likely to arise during translation, processing, or secretion. This hypothetical dysregulation of energy intake can contribute to hypertriglyceridemia and obesity seen in PD [10,30].

PD strain displays significant downregulation of three acyl-CoA-generating enzymes (ACSM3, ACSM2A, ACSL5), which can contribute to the lower ability to utilize fatty acids and hypertriglyceridemia in this strain. ACSM3 belongs to a family of enzymes conjugating medium-chain fatty acids (C4-C14) to coenzyme A. At least some members can also contribute to excretion of xenobiotics by activating carboxylic compounds (e.g., benzoate, salicylate) before conjugation with glycine [31,32]. The human genome contains seven members of the family (*ACSM1*, *2A*, *2B*, *3*, *4*, *5*, and *6*). Except *ACSM4* and *6*, all are localized to a ~400-kb cluster on chromosome 16 [10] (hg38 chr16:20409534-20797581). In the rat, the family is contained in the syntenic region on chromosome 1, including *Acsm4* but excluding *Acsm2b* since *Acsm2* is not duplicated (rn6 chr1:189241593-189541233). The function of individual family members is far from elucidated [32]. Interestingly, *Acsm3* was previously identified as a candidate gene for hypertension and other MetS traits as part of the SAH (spontaneously hypertensive rat-clone A-hypertension associated, together with *Acsm2* and *Acsm1*), but the association with hypertension was not confirmed [33,34]. The main substrates of ACSM3 enzyme are C4 fatty acids, such as butyrate and isobutyrate [35]. Butyrate can enter mitochondria for fatty acid oxidation after activation by ACSM3 or can serve as a signaling molecule regulating lipid and glucose metabolism [36]. An increased concentration of short-chain fatty acids has beneficial effects on MetS [36,37]. *Acsm3* deficiency can thus perform a protective role via putatively increasing butyrate levels. However, conversely, there can be a detrimental effect due to the relative butyrate increase with normal acetate and propionate levels. Notably, the *ob/ob* mouse microbiome has a higher proportion of butyrate over propionate and acetate producers [37]. It can be noted that the normal reaction of ACSM group transcripts in the liver after HFD seems to be upregulation, which could increase the efficiency of the liver for fat storage [38]. An absence of ACSM3 in PD can thus decrease the rate of fatty acid utilization by the liver, resulting in hyperchylomicronemia. Association studies using a large cohort (4000 subjects) representing the general population in Japan found polymorphisms in the *ACSM3* gene that were strongly associated with plasma triglyceride, plasma cholesterol, body mass index (BMI), waist-to-hip ratio (W/H), and blood pressure status. The effect of this genotype on blood pressure seemed to be conveyed through its effects on BMI and W/H [39]. Similar associations of *ACSM3* gene polymorphisms with obesity were reported by Benjafield et al. [30] and Telgmann et al. [40]. One GWAS reported an association between the *ACSM* cluster and dietary patterns [41]. Even though most of these association studies with *ACSM3* as a candidate gene used less stringent statistical criteria compared to GWAS, these results are reproducible and provide compelling evidence for *ACSM3* gene involvement in obesity and hypertriglyceridemia. In addition, in BXH/HXB recombinant inbred strains derived from BN and SHR progenitors [42], hepatic expression of the *Acsm3* gene correlates inversely with the relative weight of epididymal fat (*r* = −0.53, *p* = 0.006) and with adipocyte volume (*r* = −0.50, *p* = 0.009) [43]. None of the studies, to the best of our knowledge, reported an absence of *Acsm3* expression in the liver in connection with MetS, and *Acsm3*-deficient mice were only studied regarding hypertension [34]. Potentially interesting information about the signaling and metabolic functions of *ACSM3* comes from studies of hepatocellular carcinoma, where its downregulation is associated with poor prognosis. In tumor cells, *ACSM3* mRNA was upregulated by HNF4α, downregulated by PPAR-γ, there was a negative feedback loop between ACSM3 and AKT, and *ACSM3* expression correlated with fatty acid beta oxidation [44,45]. In addition, our previous experiments in rats that were fed a normal diet also showed zero hepatic expression of the *Acsm3* gene in the PD strain while the gene was expressed in SHR [46]. The fact that *Acsm3* is not expressed in the liver of PD rats fed both control and HFD suggests the possibility of primary genetic effects of the *Acsm3* PD variant on hypertriglyceridemia, the limitation being that we do not have a complete transcriptomic profile of unchallenged PD.

Together, these results strongly suggest that *Acsm3* may play a significant role in MetS development in the PD strain. The level of this significance, in a polygenic MetS model, such as PD, can be answered by transgenic rescue experiments in PD rats.

## 5. Conclusions

The aim of our study was a closer characterization of two rat models of MetS with distinct phenotypes and gene expression differences. We found that the PD strain was most susceptible to developing MetS after being fed a high-fat diet. Molecular pathway analysis of differentially expressed genes in the liver suggests a deficiency in lipid utilization as the major contribution to MetS development in PD, with a loss of function of *Acsm3* as the most relevant candidate gene for this intrinsically polygenic model.

## Figures and Tables

**Figure 1 nutrients-13-01462-f001:**
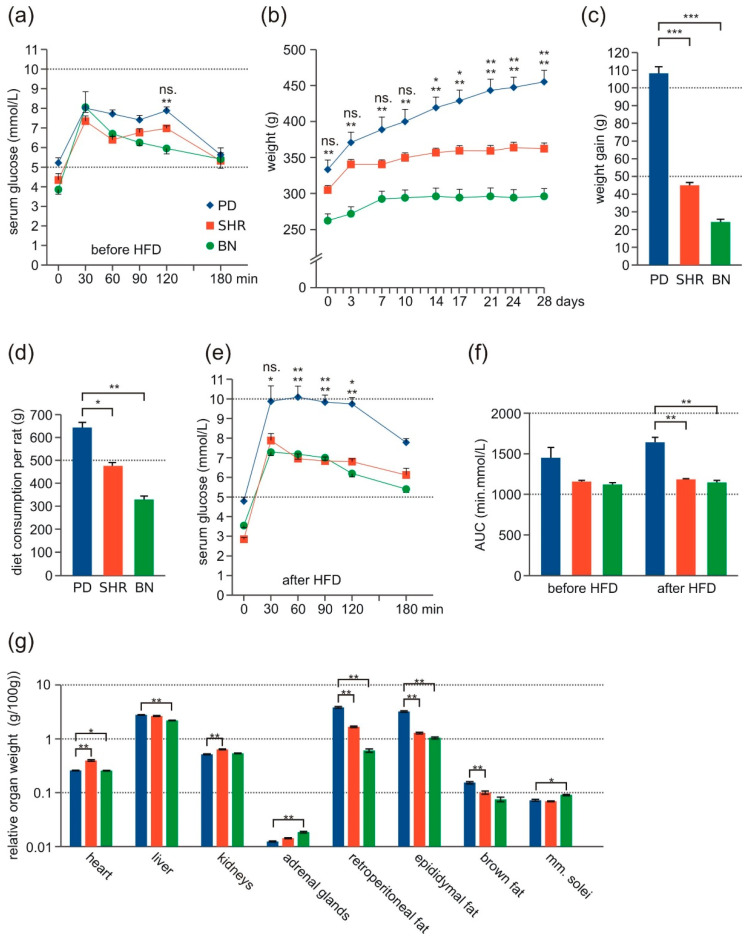
Glucose tolerance and morphometric profile of adult male rats-PD (blue, *n* = 7), SHR (red, *n* = 6), and BN (green, *n* = 7–8). (**a**) Oral glucose tolerance test (OGTT)-time course of glucose level before HFD. (**b**) Time course of body weight during HFD feeding. (**c**) Absolute weight gain during HFD feeding. (**d**) Consumed diet amount per rat of each strain during HFD feeding. (**e**) OGTT-time course of glucose level after HFD. (**f**) The area under the curve (AUC) of OGTT before and after HFD. (**g**) Relative tissue weights per 100 g body weight after HFD. Data are presented as arithmetic means ± SEM. Statistical significance levels for the factor strain of repeated measurements ANOVA (weight, OGTT), one-way ANOVA (weight gain, consumed diet, tissue weight) or two-way ANOVA (AUC) are indicated for pair-wise post hoc Tukey’s test as follows: * *p* < 0.05, ** *p* < 0.01, *** *p* < 0.001. For post-hoc OGTT and weight, the upper symbol corresponds to PD vs. SHR comparison, the bottom symbol represents PD vs. BN comparison, ns. = not significant.

**Figure 2 nutrients-13-01462-f002:**
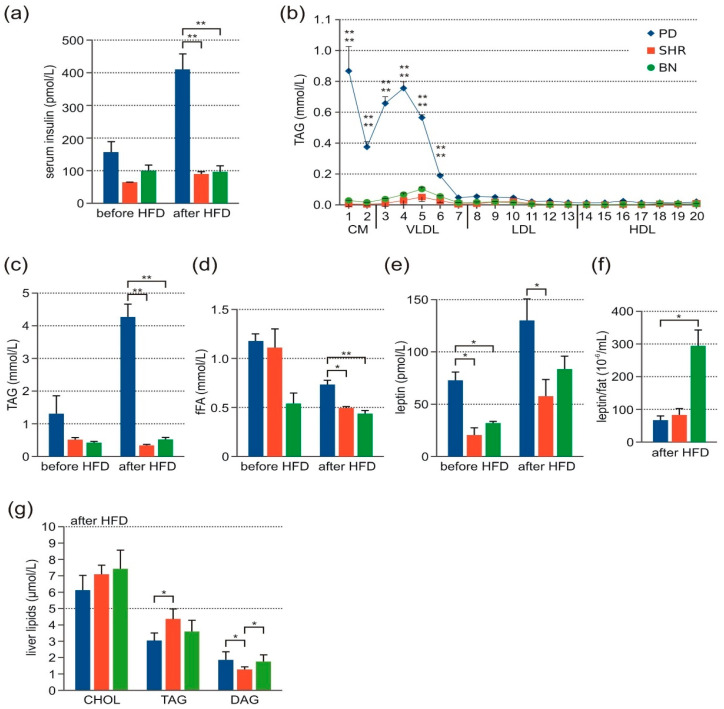
Laboratory findings in adult male rats-PD (blue, *n* = 3–7), SHR (red, *n* = 3–6), and BN (green, *n* = 3–8). (**a**) Insulin levels before and after HFD. (**b**) TAG levels after HFD in 20 lipoprotein fractions. (**c**) Total TAG levels before and after HFD. (**d**) Free FAs before and after HFD. (**e**) Absolute leptin blood level before and after HFD. (**f**) Relative leptin blood level (after correction to fat tissue amount) after HFD. (**g**) Cholesterol, TAG, and DAG levels in liver tissue. Data are presented as arithmetic means ± SEM. Statistical significance levels for the factor strain of repeated measurements ANOVA (cholesterol and TAG in 20 lipoprotein fractions) or one-way ANOVA (insulin levels, total TAG, fFAs, leptin, cholesterol, TAG, and DAG in liver tissue) are indicated for pair-wise post hoc Tukey’s test as follows: * *p* < 0.05, ** *p* < 0.01. For post-hoc TAG levels, the upper symbol corresponds to PD vs. SHR comparison, the bottom symbol represents PD vs. BN comparison.

**Figure 3 nutrients-13-01462-f003:**
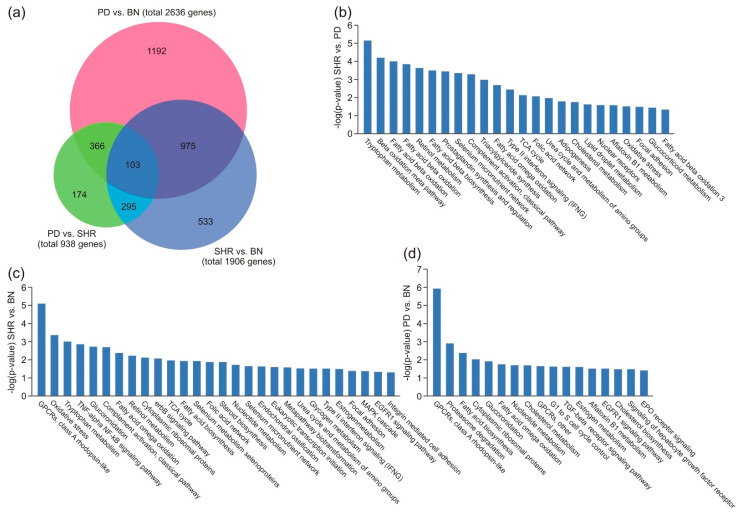
Differentially expressed genes in the liver after HFD. Venn diagram (**a**), pathway enrichment analysis using Transcriptomic analysis console software (**b**–**d**). (**b**) Pathway enrichment analysis-SHR vs. PD comparison. (**c**) Pathway enrichment analysis—SHR vs. BN comparison. (**d**) Pathway enrichment analysis − PD vs. BN comparison.

**Figure 4 nutrients-13-01462-f004:**
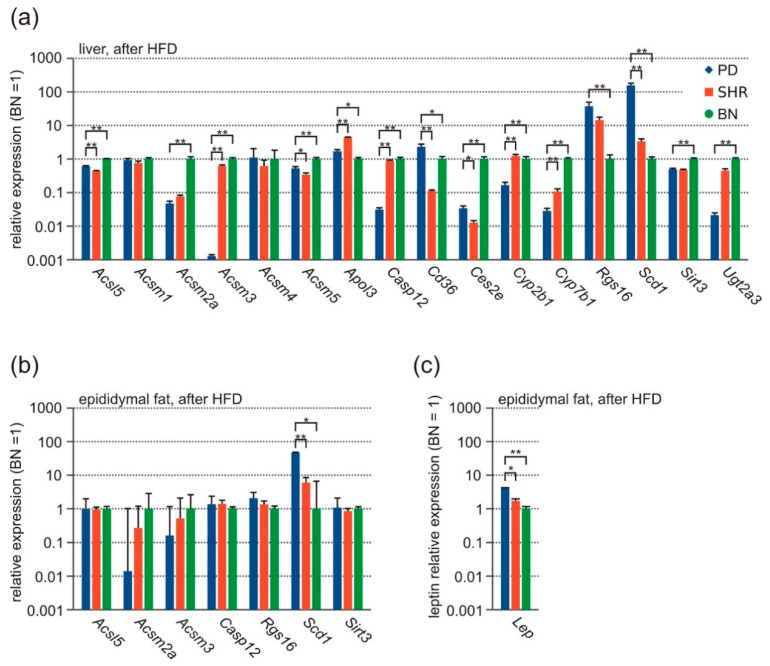
Gene expression (qPCR) after HFD in adult male rats-PD (blue, *n* = 7), SHR (red, *n* = 6), and BN (green, *n* = 8). (**a**) Liver tissue. (**b**) Epididymal fat tissue. (**c**) Leptin expression in epididymal fat. Data are presented as arithmetic means ± SEM. Statistical significance levels for the factor strain of one-way ANOVA are indicated for pair-wise post hoc Tukey’s test as follows: * *p* < 0.05, ** *p* < 0.01.

**Figure 5 nutrients-13-01462-f005:**
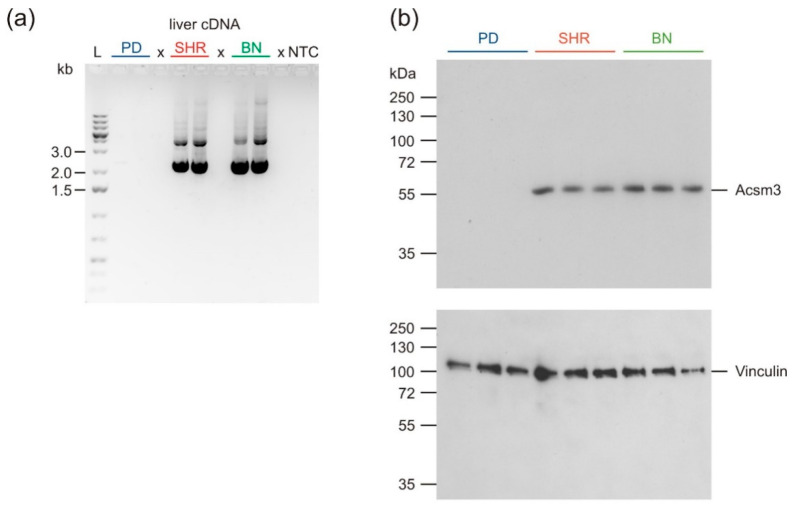
Expression of *Acsm3* in the liver of adult male rats-PD (blue), SHR (red), and BN (green); after HFD. (**a**) RT-PCR of *Acsm3*, visualized on gel electrophoresis (two representative samples for each strain). Primers flank the complete coding sequence; the expected size of the PCR product is 2296 bp. (**b**) Western blotting of ACSM3 (expected molecular weight of the mature polypeptide 63.2 kDa). Vinculin (expected molecular weight 116.6 kDa) was used as the loading control (three representative samples for each strain).

## Data Availability

The microarray data discussed in this publication have been deposited in NCBI’s Gene Expression Omnibus [16] and are accessible through GEO Series accession number GSE164176.

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
