# Peer review of "Hepatic Transcriptome Profiling Reveals Lack of Acsm3 Expression in Polydactylous Rats with High-Fat Diet-Induced Hypertriglyceridemia and Visceral Fat Accumulation"

_nutrients, 2021, doi:10.3390/nu13051462_

Round 1

Reviewer 1 Report

The authors have shown data from experimental animal studies from which they suggest disturbances in fatty acid utilization as a molecular mechanism predisposing an animal model of hypertriglyceridemia, insulin resistance and obesity to hypertriglyceridemia and fat accumulation, since they did not express Acsm3 (acyl-CoA synthetase medium-chain family member) gene. These findings were brought from analysis of liver transcriptome revealing dysregulation of multiple pathways contributing for that, with concomitant decreased of glucose tolerance and hyperinsulinemia. The study design seemed to be appropriated since they used other 2 animal models for comparisons of the effect of a HFD for 4 weeks.

Well design, properly analyzed, results were well presented and the language was clear and easy to follow. The results were properly discussed.

Author Response

We thank the reviewer for the positive, encouraging opinion about our work.

Reviewer 2 Report

Junková et al., investigate and characterize metabolic syndrome in the Polydactylous (PD) rat model relative to the Spontaneously hypertensive rat and Brown Norway rat strains, ultimately showing that Acsm3 is not expressed the livers of PD rats fed a high fat diet (HFD).  The authors used many assays to metabolic parameters to characterize the rats before being fed a HFD and after a HFD to demonstrate the onset of metabolic syndrome (MetS).  They also include hepatic transcriptomic data and pathway analysis to compare pathways affected by a HFD and verify the gene expressions by RT-qPCR.  The manuscript has promise, but there are several concerns that need to be addressed.

Major

  1. Introduction, lines 47-50: This statement is long, confusing, and took multiple times reading to understand the complete construct.  Please break down into shorter sentences and clarify.
  2. Introduction: The explanations regarding GWAS need to be tied into the paper as the title and abstract indicate this is a transcriptomic and metabolic paper.
  3. Introduction, lines 54-69: Split into at least two separate paragraphs.  One paragraph to describing the literature regarding the rat strains used in this study and MetS and why they are good models to use for GWAS transcriptomic studies. Second paragraph with the gap in literature and why the current study expands the MetS field.
  4. Introduction: The title and a key purpose of the paper is the lack of expression of Acsm3 in the liver of PD mice.  A paragraph is needed describing gene expression in the liver of MetS rodent models and why Acsm3 is important in MetS.  
  5. Results lines 277-283: Create a 3-way Venn diagram showing the comparisons of differentially expressed genes between the mice (i.e. the number of genes differentially expressed [DE] in all the strains, the ones DE by two of the strains but not the other, and the genes DE by an individual strain).  Also, report how many genes were expressed in total for each strain; strongly recommended to include a Venn diagram for the number of expressed genes as described above.
  6. Results lines 328-333: I’m not convinced the sequencing of Acsm3 is necessary.  The expression was identified in the fat and the PD rat model, as described in the manuscript, and does not indicate that mutations would occur in a single organ.
  7. Discussion lines 344-351: This explanation at the end of the introduction.  I was lost most of the manuscript as why these rat strains were selected.
  8. Discussion: Acsm3 is discussed and concluded to be a possible contributor to MetS in the PD rat model. A fairly well detailed discussion of Acsm3 is also included in relation to health conditions in humans and mice.  However, I think there needs to be further discussion as to the function of the Acsm gene family (specifically Acsm3) in a health population and a discussion of Acsm3 in an unhealthy state.  How novel is it for there to be no Acsm3 expression in any metabolic disease model (MetS, obesity, diabetes, etc.)?
  9. General: The biggest concern is a lack of control within a rat stain that matches the HFD fed group.  In particular, there are no transcriptomic studies in PD rats.  Do PD rats on normal chow express Acsm3 and what genes are DE within a strain between normal chow and HFD.  Because there is no normal chow control group, I question the gene enrichment analysis (Results lines 285-295) since the gene selection criteria was for DE genes between rats.  In addition, most transcriptomic studies in animal models have a baseline gene expression for comparison, so the results of this study and how genes are expressed in a treated PD rat are difficult to interpret without a baseline comparison of an untreated PD rat.  This leaves a lot of potential for the PD rat to be used as a model for metabolic disease, but the normal leaves of transcription need to be characterized. 

Minor

  1. Results line 335: It’s very good to have the Western blot.  Although the figure is self-explanatory in nature, the text needs a better description than its current state.
  2. Discussion lines 386-388:   Just state that there was no immune activation indicated by the cytokine levels. 

Author Response

We would like to thank the reviewer for insightful comments. Thus we were able to improve the manuscript substantially, as detailed below, where we respond to the reviewer´s comments.

Junková et al., investigate and characterize metabolic syndrome in the Polydactylous (PD) rat model relative to the Spontaneously hypertensive rat and Brown Norway rat strains, ultimately showing that Acsm3 is not expressed the livers of PD rats fed a high fat diet (HFD).  The authors used many assays to metabolic parameters to characterize the rats before being fed a HFD and after a HFD to demonstrate the onset of metabolic syndrome (MetS).  They also include hepatic transcriptomic data and pathway analysis to compare pathways affected by a HFD and verify the gene expressions by RT-qPCR.  The manuscript has promise, but there are several concerns that need to be addressed.

Major

  1. Introduction, lines 47-50: This statement is long, confusing, and took multiple times reading to understand the complete construct.  Please break down into shorter sentences and clarify.

Response:

We thank the reviewer and the revised Introduction is shorter and more focused on metabolic analysis of PD strain.

2. Introduction: The explanations regarding GWAS need to be tied into the paper as the title and abstract indicate this is a transcriptomic and metabolic paper.

Response:

We mentioned problems of GWAS in humans and the advantages of animal models to provide information about molecular mechanisms underlying metabolic syndrome.

3. Introduction, lines 54-69: Split into at least two separate paragraphs.  One paragraph to describing the literature regarding the rat strains used in this study and MetS and why they are good models to use for GWAS transcriptomic studies. Second paragraph with the gap in literature and why the current study expands the MetS field.

Response:

Rat strains in the current study were not used “for GWAS transcriptomic studies”, i.e. for identification of statistical associations of genotypes with phenotypes. In the current study, we compared deregulated genes between 3 inbred strains after treatment with HFD. We modified the paragraph in question as follows: The first part was removed as it was redundant with the paragraph transferred from Discussion (see point 7).

4. Introduction: The title and a key purpose of the paper is the lack of expression of Acsm3 in the liver of PD mice.  A paragraph is needed describing gene expression in the liver of MetS rodent models and why Acsm3 is important in MetS.  

Response:

We included a short information about Acsm3 to the Introduction. However, as the functional consequences of Acsm3 can be quite complicated, and most are still hypothetical, we keep this in Discussion.

5. Results lines 277-283: Create a 3-way Venn diagram showing the comparisons of differentially expressed genes between the mice (i.e. the number of genes differentially expressed [DE] in all the strains, the ones DE by two of the strains but not the other, and the genes DE by an individual strain).  Also, report how many genes were expressed in total for each strain; strongly recommended to include a Venn diagram for the number of expressed genes as described above.

Response:

We included Venn diagram of differentially expressed genes as Figure 3a. We modified Methods to include the way the Venn diagram was produced and modified Results to include reference to figure 3a. Original figure 3a-c is 3b-d in the revised manuscript.

6. Results lines 328-333: I’m not convinced the sequencing of Acsm3 is necessary.  The expression was identified in the fat and the PD rat model, as described in the manuscript, and does not indicate that mutations would occur in a single organ.

Response:

Lack of Acsm3 expression in liver and detectable mRNA expression in fat may indicate a mutation in a promoter/enhancer region specific for liver expression. This intention was not clear from the manuscript, therefore the Results, lines 329-330 were modified as follows: “Absent Acsm3 expression in liver of PD strain compared to SHR and BN strains (Figure 5a) suggests a sequence variant in a tissue specific cis-regulatory element (e.g. promoter, enhancer, differential splice site) precluding liver expression.”

7. Discussion lines 344-351: This explanation at the end of the introduction.  I was lost most of the manuscript as why these rat strains were selected.

Response:

This section of the discussion was transferred to introduction accordingly, as a second to last paragraph (see also point 3).

8. Discussion: Acsm3 is discussed and concluded to be a possible contributor to MetS in the PD rat model. A fairly well detailed discussion of Acsm3 is also included in relation to health conditions in humans and mice.  However, I think there needs to be further discussion as to the function of the Acsm gene family (specifically Acsm3) in a health population and a discussion of Acsm3 in an unhealthy state.  How novel is it for there to be no Acsm3 expression in any metabolic disease model (MetS, obesity, diabetes, etc.)?

Response:

We thank the reviewer for this comment. We expanded Discussion to provide detail about Ascm family and Acsm3 in particular. Unfortunately the information that is available for Acsm3 and MetS disease state is limited to association studies that do not provide functional data. Also, Acsm3 knock-out in the mouse was only studied regarding hypertension (which was not influenced). Therefore Acsm3 lack in an animal strain with MetS seems to be indeed novel.

9. General: The biggest concern is a lack of control within a rat stain that matches the HFD fed group.  In particular, there are no transcriptomic studies in PD rats.  Do PD rats on normal chow express Acsm3 and what genes are DE within a strain between normal chow and HFD.  Because there is no normal chow control group, I question the gene enrichment analysis (Results lines 285-295) since the gene selection criteria was for DE genes between rats.  In addition, most transcriptomic studies in animal models have a baseline gene expression for comparison, so the results of this study and how genes are expressed in a treated PD rat are difficult to interpret without a baseline comparison of an untreated PD rat.  This leaves a lot of potential for the PD rat to be used as a model for metabolic disease, but the normal leaves of transcription need to be characterized. 

Response:

We agree with the reviewer that it would be interesting to have gene expression profiles in all strains on control diet and HFD and to search for strain x diet interactions. On the other hand, we observed practically no significant differences between strains on normal control diet. Accordingly, differences in gene expression on normal diet would not be informative about genetic predisposition of strains to e.g. hypertriglyceridemia. In addition, our previous experiments in rats that were fed normal diet also showed zero hepatic expression of Acsm3 gene in PD strain while the gene was expressed in the SHR. The fact that Acsm3 gene is not expressed in the liver of PD rats fed both control and HFD suggests the possibility of primary genetic effects of Acsm3 PD variant on hypertriglyceridemia. This information was added to the revised manuscript.

Minor

  1. Results line 335: It’s very good to have the Western blot.  Although the figure is self-explanatory in nature, the text needs a better description than its current state.

Response:

We modified the text to include more information as follows: “Absent Acsm3 liver expression in PD on the mRNA level prompted us to test expression of the protein. Western blot analysis of liver tissue lysates proved the absence of ACSM3 protein in PD (Figure 5b).”

2. Discussion lines 386-388:   Just state that there was no immune activation indicated by the cytokine levels.

Response:

The revised sentence now runs: “However, there was no immune activation in PD as indicated by the serum cytokine levels.”
